# Roles of Androgen Receptor Signaling in Urothelial Carcinoma

**DOI:** 10.3390/cancers16040746

**Published:** 2024-02-10

**Authors:** Debasish Sundi, Katharine A. Collier, Yuanquan Yang, Dayssy Alexandra Diaz, Kamal S. Pohar, Eric A. Singer, Sanjay Gupta, William E. Carson, Steven K. Clinton, Zihai Li, Edward M. Messing

**Affiliations:** 1Department of Urology, Division of Urologic Oncology, Pelotonia Institute for Immuno-Oncology, The Ohio State University Wexner Medical Center, Columbus, OH 43210, USA; 2Department of Internal Medicine, Division of Medical Oncology, Pelotonia Institute for Immuno-Oncology, The Ohio State University Wexner Medical Center, Columbus, OH 43210, USA; 3Department of Radiation Oncology, The Ohio State University Wexner Medical Center, Columbus, OH 43210, USA; 4Department of Urology, Division of Urologic Oncology, The Ohio State University Wexner Medical Center, Columbus, OH 43210, USAeric.singer@osumc.edu (E.A.S.); 5Department of Urology, Case Western Reserve University School of Medicine, The Urology Institute, University Hospitals Cleveland Medical Center, Cleveland, OH 44106, USA; gxs44@case.edu; 6Department of Surgery, Division of Surgical Oncology, Pelotonia Institute for Immuno-Oncology, The Ohio State University Wexner Medical Center, Columbus, OH 43210, USA; 7Department of Internal Medicine, Division of Medical Oncology, The Ohio State University Wexner Medical Center, Columbus, OH 43210, USA; 8Departments of Urology, Oncology, and Pathology, University of Rochester Medical Center, Rochester, NY 14642, USA

**Keywords:** bladder cancer, androgen receptor, anti-tumor immunity

## Abstract

**Simple Summary:**

Androgen receptor signaling is an important target to investigate in urothelial carcinoma of the bladder because basic and clinical evidence strongly suggests that the androgen receptor is involved in bladder carcinogenesis. Moreover, androgen receptor signaling may suppress anti-tumor immunity. These factors may also be important contributors to the strong male sex bias in urothelial carcinoma of the bladder.

**Abstract:**

Preclinical and clinical data suggest that androgen receptor signaling strongly contributes to bladder cancer development. The roles of the androgen receptor in bladder carcinogenesis have obvious implications for understanding the strong male sex bias in this disease and for potential therapeutic strategies as well. In this review, we summarize what is known about androgen receptor signaling in urothelial carcinoma as well as in tumor-infiltrating immune cells, reviewing preclinical and clinical data. We also highlight clinical trial efforts in this area.

## 1. Introduction

Bladder cancer is a relatively common solid tumor with a global burden that is amplified by high mortality rates [1,2]. Intense/lifelong treatment/monitoring requirements lead patients with bladder cancer to incur the highest treatment costs among all cancer types [3]. Bladder cancer also features a stark sex bias, with 75–80% of incident cases occurring in men [1,4]. This male sex bias is not fully accounted for by established risk factors such as smoking [5]. Thus, biological sex differences that are related to bladder cancer risk have emerged as important topics of study to not only understand bladder cancer pathogenesis better but also define novel and more effective therapeutic targets. Foremost among biological sex differences are the pleiotropic effects of sex hormone signaling, including those mediated by androgens and androgen receptors (ARs). Dating back to the seminal work of Huggins and Hodges [6,7], who demonstrated androgens to be drivers of neoplastic prostate cells, targeting AR has been a backbone of prostate cancer therapy. Owing to a growing and compelling body of work linking the AR to bladder cancer, anti-androgens may play an integral role in the next generation of bladder cancer therapies.

## 2. AR Expression in Bladder Cancer

While adenocarcinoma of the prostate is canonically associated with AR positivity, urothelial carcinoma of the bladder also expresses the AR protein, as demonstrated by several early studies [8,9,10,11]. Subsequent research into how immunohistochemical AR positivity is associated with bladder cancer stage generally suggests that low-grade [12] and earlier-stage [13,14] bladder cancers are more likely to express the AR. The majority (75%) of newly diagnosed cases are non-muscle-invasive bladder cancer (NMIBC), comprising clinical stages Ta, Tis, and T1. AR positivity within NMIBC cohorts ranges from 33 to 75% [15,16,17,18,19], with one study interestingly demonstrating equivalent rates of AR positivity between male and female patients [17]. Given the association of AR with bladder cancer, Wu et al. used a whole-exome sequencing approach to analyze a 99-patient cohort and did not detect any somatic AR mutations in any of the patients, which suggests that while the AR is often expressed in bladder cancer, AR mutation is not a key event driving bladder carcinogenesis [20]. The absence of AR mutations and amplifications in bladder cancer clinical cohorts was later confirmed by Necchi et al. [21].

## 3. AR and Bladder Cancer Development in Preclinical Models

Several key studies have shown that AR signaling is required for bladder tumor development in different carcinogen-induced orthotopic murine models [22]. N-butyl-N-(4-hydroxybutyl)nitrosamine (BBN) is a carcinogen found in cigarette smoke that specifically induces urothelial bladder cancers in wild-type murine subjects, and these tumors closely mimic human bladder cancers both histologically and molecularly [23,24].

Imada et al. exposed male Wistar rats to BBN and found that multiple forms of ADT (surgical castration, flutamide, or LH-RH agonist) all decreased bladder cancer incidence [25]. Subsequently, Miyamoto et al. performed the defining study in this field by examining BBN-induced bladder carcinogenesis in male and female C57BL/6 mice, comparing wild-type subjects to AR knockouts (ARKOs) [26]. In this study, bladder cancer rates after exposure to 0.05% BBN were higher in wild-type males than in females (92% vs. 42%, n = 12 mice per group, *p* = 0.0136). No tumors developed in ARKO subjects of either sex. Exogenous testosterone administered to ARKO male subjects, interestingly, appeared to rescue tumor formation in 25% of subjects. To corroborate their findings, Miyamoto et al. assessed *in vitro* proliferation and *in vivo* xenograft growth of two AR-expressing human urothelial carcinoma cells lines (UMUC-3 and TCC-SUP). *In vitro*, supplementation with the potent androgen dihydrotestosterone increased cancer cell proliferation; *in vivo*, castration or anti-androgen therapies decreased xenograft size and weights. Kameyama et al. studied parental and gemcitabine-resistant T24 urothelial carcinoma cell lines and found that the gemcitabine-resistant cells overexpressed AR at the mRNA and protein levels and had reduced proliferation with enzalutamide treatment [27].

Johnson A.M. et al. studied a conditional transgenic mouse model (UPII-SV40T) with urothelial-specific expression of the SV40 T antigen [28]. In this model, consistent with the prior study by Miyamoto et al., surgical castration decreased bladder tumor volumes (which could be rescued with exogenous testosterone). A follow-up study by Hsu et al. evaluated BBN-induced tumor formation in urothelial AR knockout (Uro-AR^−/y^) subjects compared to wild-type controls and showed that Uro-AR null subjects had a significantly lower incidence of bladder cancer [29]. Further confirming the causality of AR signaling with bladder carcinogenesis, D.T. Johnson et al. showed that conditional over-expression of a human AR transgene increased murine bladder tumor incidence with BBN exposure in both male and female subjects [30].

In the murine background, AR has been shown to influence the activity of the anti-angiogenic signal thrombospondin-I [28], UDP-glucoronosyltransferases that metabolize/detoxify carcinogens [31], the epithelial-to-mesenchymal transition molecule CD24 [32,33,34], VEGF [33], ERBB2 [35], and EGFR [35,36,37]. Zhang et al. demonstrated that surgical castration led to increased survival among murine subjects with BBN-induced bladder cancers [38]. Shang et al. performed an intriguing study in the BBN-model, showing that the anti-tumor effect of bacille Calmette–Guérin (BCG, an immunotherapy frequently used in bladder cancer [39]) was augmented using multiple AR-targeting strategies, and it also increased bladder infiltration by F4/80^+^ macrophages [40].

Taken together, preclinical studies of AR and urothelial bladder cancer to date have strongly established a causal effect of AR signaling that can be clinically leveraged for therapy. 

## 4. AR and Bladder Cancer Risk in Clinical Cohorts

Multiple observational studies suggest that reduced AR signaling is linked to lower bladder cancer incidence. Izumi et al. analyzed a cohort of 162 male patients with prostate cancer and bladder cancer. Androgen deprivation therapy (ADT) was strongly associated with prevention of bladder cancer recurrence [41]. In a closely related study, Izumi et al. found that immunohistochemical AR positivity among patients treated with ADT was associated with increased time to bladder cancer recurrence, thus implicating AR as a predictive biomarker for ADT in bladder cancer [42]. Studying a 72,000-patient cohort of men in the Prostate, Lung, Colorectal, and Ovarian (PLCO) screening trial, Morales et al. found that use of the oral 5-alpha reductase inhibitor finasteride decreased bladder cancer risk over time, even after adjusting for patient age and tobacco use [43]. Specifically, the self-reported use of finasteride was associated with a hazard ratio of 0.634 (95% confidence interval 0.493–0.816) compared to non-use. Sathianathen et al. could not confirm the chemopreventive effect of finasteride, however, when conducting a secondary analysis of men in the Medical Therapy for Prostatic Symptoms (MTOPS) trial. It should be noted that the MTOPS trial had a much smaller cohort, n = 2700, than PLCO) [44]. A notable finding of the secondary analysis of MTOPS performed by Sathianathen et al., however, was that it confirmed lower serum dihydrotestosterone (DHT) levels in patients taking finasteride compared with non-users. Several subsequent reports have suggested that among patients with non-muscle-invasive bladder cancer, therapies that target androgen signaling are associated with a lower rate of bladder cancer recurrence [45,46,47]. Taken together, most observational clinical data are consistent with preclinical research that links AR signaling to bladder cancer risk.

## 5. AR Regulation of Anti-Tumor Immunity

Several key studies suggest that targeting androgen signaling can improve anti-tumor immunity in bladder cancer. Guan et al. demonstrated in prostate cancer that androgen signaling impairs CD8^+^ T-cell-mediated anti-tumor immunity in the setting of anti-PD-1 checkpoint blockade [48]. In this study, AR was proposed to repress IFNγ gene expression in CD8^+^ T cells. In multiple bladder cancer preclinical models, Kwon et al. found that the male sex bias of bladder cancer is largely mediated by CD8^+^ T cells, that AR signaling promotes CD8^+^ T-cell exhaustion via TCF1, and that a blockade of androgen signaling improves the efficacy of the anti-PD-1 checkpoint blockade [49]. Specifically, this work showed that bladder tumor growth *in vivo* was faster in men than in women, and this was mediated specifically by CD8^+^ T cells (established through lymphodepletion experiments in C57BL/6 mice). Among tumor-infiltrating leukocytes (TILs), CD8^+^ cells from women produced cytotoxic effector molecules IFNγ and GZMB in greater quantities. In fact, adoptive transfer of female T cells into tumor-bearing T-cell-deficient subjects eliminated the sex-based disparity in bladder tumor growth. Single-cell genomic analysis of murine bladder tumors pointed to TCF1 having AR as a target, which was an association that was thoroughly evaluated by the study group as well. This work by Kwon et al. definitely established the roles of T-cell intrinsic AR in driving CD8^+^ T-cell exhaustion and shall have far reaching implications beyond bladder cancer. 

Subsequently, Besancon et al. showed in the MBT-2 murine bladder cancer model that the anti-androgen enzalutamide improved responses to anti-PD-1 immunotherapy even though enzalutamide monotherapy did not appear to alter the tumor immune microenvironment when assayed for T, dendritic, and myeloid-derived suppressor cells [50]. Conversely, tumor-infiltrating B [51] and T [52] cells may in fact induce AR signaling in bladder cancer cells, which is particularly relevant given recent studies pertaining to the potential of tertiary lymphoid structures to determine prognosis and immunotherapy responses in bladder cancer and other tumor types [53,54,55,56]. Due to these studies’ findings, it is quite likely that impaired adaptive immune responses underlie at least part of the sex disparity in bladder cancer and that targeting androgen signaling may improve clinical responses to checkpoint blockade immunotherapy. Figure 1 illustrates the predicted net direct and indirect (immune-mediated) effects of AR signaling on bladder cancer based on the above findings.

## 6. Mechanisms of AR Signaling in Bladder Cancer

The potentially profound effects of androgens and AR signaling on bladder cancer have prompted many investigators to study mechanistic links to the AR within bladder cancer models. Chen et al. studied the effect of BCG and the potent androgen dihydrotestosterone (DHT) on two bladder cancer cell lines. The study found that BCG-induced bladder cancer cell lines led to the upregulation of NF-κB and IL-6, an effect that was abrogated by DHT [57]. This finding is clinically relevant as IL-6 promotes BCG binding to cancer cells, thus suggesting that androgen signaling may impair responses to the most commonly used immunotherapy in bladder cancer. Consistent with this concept, more recent evidence shows that the AR degradation enhancer ASC-J9 or the anti-androgen flutamide enhances the anti-tumor effects of BCG immunotherapy in bladder cancer *in vitro* and *in vivo* [40]; it also shows a possible mechanism of AR-mediated BCG resistance through the GTPase Rab27b [58].

The research group led by Miyamoto found that DHT signaling via ARs upregulated EGFR and ERBB2 oncogenes [35], ELK1 [59], β-catenin [60], and ATF2 [61] in multiple human bladder cancer cell lines; downregulated levels of UDP-glucoronosyltransferases, a class of molecules that metabolizes and detoxifies carcinogens [31]; and downregulated the tumor suppressor FOXO1 [62]. In addition, the same group found that EGFR, while induced by AR signaling, also leads at AR upregulation [36]. Further strengthening the interplay between AR and EGFR, Hsieh et al. reported that EGF can in fact transactivate AR, even at very low androgen levels [37].

Additional reported mechanistic associations of AR in bladder cancer suggest that AR signaling is truly pluripotent in this disease. Overdevest et al. found that the transmembrane protein CD24, which mediates cell proliferation in several cancer types, is partially responsible for carcinogenesis in the BBN model (especially in male subjects). Furthermore, Overdevest et al. observed that CD24 is upregulated in the presence of androgens [32]. The association of AR and CD24 has since been independently confirmed *in vitro* [33] and may relate to the transcriptional regulators GON4L and YY1 [34]. Multiple groups have reported that androgen signaling upregulates bladder cancer cell proliferation through canonical epithelial–mesenchymal transition pathway members [63,64]. AR signaling may also have anti-tumor effects: Sottnik et al. recently showed that AR signaling represses CD44, which is linked to tumor progression in different cancer types [65]. 

A visual synthesis of potential tumor-intrinsic and immune-mediated effects of androgen receptor signaling in bladder cancer is illustrated in Figure 2.

## 7. AR Signaling as a Therapeutic Target in Clinical Trials

Owing to a growing body of work that strongly suggests AR signaling to be a therapeutic vulnerability in bladder cancer, several recent and upcoming trials will evaluate anti-androgen treatments in prospective clinical cohorts. For example, both clinical and preclinical studies suggest that AR signaling is a reversible cause of therapeutic resistance to multiple bladder-cancer-directed therapies including BCG immunotherapy [58], radiation [66], and cisplatin chemotherapy [67,68,69,70]. A clinicaltrials.gov query for studies pertaining to “bladder cancer” and “androgen” revealed six relevant recent or ongoing trials, which are summarized in Table 1. 

In the Phase I/Ib Trial of Enzalutamide and Gemcitabine and Cisplatin in Metastatic Bladder Cancer (NCT02300610), sponsored by Moffitt Cancer Center, 10 patients were evaluated in dose escalation and expansion cohorts, and no dose-limiting toxicities to the regimen were found [71]. Interestingly, one complete responder in this trial was a female patient with strong tumor AR expression. This result suggests the potential for anti-AR therapeutics to extend to both sexes, rather than only applying to male patients with bladder cancer.

Enzalutamide for Bladder Cancer Chemoprevention (NCT02605863) was a phase II trial sponsored by the University of Rochester that studied the oral anti-androgen enzalutamide as a strategy to prevent recurrences in patients with low- or intermediate-risk non-muscle-invasive bladder cancer. Though promising, this study was ultimately withdrawn due to a lower-than-expected accrual rate. This study was the first protocol to evaluate anti-androgens in the setting of bladder cancer chemoprevention.

Influence of Hormone Treatment in Radiation Therapy for Bladder Cancer (NCT04282876) is a 60-patient observational study sponsored by Aarhus University Hospital focused on the gonadotropin-releasing hormone antagonist degarelix and bladder fibrosis in a cohort of patients undergoing pelvic radiation therapy for cT2-4 bladder cancer. Bladder fibrosis, if related to myofibroblast enrichment in the tumor, may be associated with luminal subtype bladder cancers that are also highly infiltrated with immune cells [72], which suggests that this study will have implications for combining anti-androgens with immunotherapy in bladder cancer. 

The Phase II Trial of Bicalutamide in Patients Receiving Intravesical BCG for Non-muscle Invasive Bladder Cancer (BicaBCa) (NCT05327647) is a two-arm study sponsored by CHU de Québec-Université Laval evaluating time to recurrence among patients receiving standard of care (intravesical bacille Calmette–Guérin, BCG) compared to BCG plus the oral anti-androgen bicalutamide. If bicalutamide can alter the balance of effector and exhausted T cells in this study, one might expect to observe a longer time to recurrence among patients in the combination treatment arm.

Effects of Apalutamide on EGFR Expression in Patients with Non-muscle Invasive Bladder Cancer (NCT05521698) is a Phase II study sponsored by the U.S. National Cancer Institute that will randomize patients suspected of harboring non-muscle-invasive bladder cancer based on cystoscopy to standard of care (transurethral resection of bladder tumor, TURBT) compared to the oral anti-androgen apalutamide in the window of opportunity prior to TURBT. The correlative outcomes in this trial will test the hypotheses that apalutamide will decrease EGFR activity and will alter the tumor microenvironment in bladder cancer, including T-cell functional markers. This study is the second protocol to evaluate anti-androgens for bladder cancer in the setting of chemoprevention, and it includes six North American centers.

Targeting Androgen Signaling in Urothelial Cell Carcinoma—Neoadjuvant (TASUC-Neo) is a phase I study sponsored by Brown University that will study 32 patients with localized or locally advanced (N0-1) muscle-invasive bladder cancer (cT2-4). This study will combine standard of care neoadjuvant gemcitabine–cisplatin chemotherapy with the gonadotropin-releasing hormone antagonist degarelix if the participants’ tumors are considered AR-expressing via immunohistochemistry. The primary outcome of this trial is rate of pathologic complete response (pT0). This protocol is novel by way of evaluating the potential of anti-androgen therapy with a conventional urothelial-cancer-directed chemotherapy.

All of the existing protocols are early-phase studies designed around the principle that AR signaling is an important driver of bladder urothelial carcinogenesis. Correlative studies may help with predictive biomarker development and/or reveal optimal combination strategies with anti-androgen therapies.

## 8. Bladder Cancer Sex Bias: Alternative Explanatory Factors

While AR signaling plays an important role in the substantial male sex bias observed in bladder cancer, it is important to acknowledge non-biological factors (e.g., social, behavioral, occupational, environmental) and biological explanations other than the androgen receptor as well. 

Social and environmental variables: In a systematic review and meta-analysis performed by Cumberbatch et al., distinct occupational exposures were associated with bladder cancer incidence: the highest risks were borne by “workers exposed to aromatic amines (tobacco, dye, and rubber workers; hairdressers; printers; and leather workers) and polyclinic aromatic hydrocarbons (PAHs) (chimney sweeps, nurses and waiters, aluminum workers, seamen, and oil/petroleum workers)” [73]. The extent to which these occupations are enriched for male or female participants may, in turn, influence bladder cancer sex bias. Tobacco smoking is a leading behavioral risk factor for bladder cancer. Globally, smoking (in various forms such as cigarettes, cigars, or vapes) is more common in men than in women [74,75,76,77]. However, a case-control analysis of bladder cancer risk factors conducted by Hartge et al. showed that occupational, environmental, and behavioral risk factors could not explain the male sex bias in bladder cancer; after adjustment for explanatory factors, men still bore an elevated incidence ratio of bladder cancer (2.7, referent women) [5]. While a comprehensive study of environmental-exposure-related risk factors for bladder cancer is beyond the scope of this review, it is important to acknowledge that such factors contribute to some but not all of the sex bias observed in this disease.

Y chromosome loss: Mosaic loss of the Y chromosome (LOY) occurs in male tissues with ageing and predisposes men to cancers and benign diseases [78]. Seminal work from Abdel-Hafez, Schafer, Chen et al. studied LOY in bladder cancer *in vitro*, *in vivo*, and in clinical datasets [79]. While LOY did not impact bladder cancer cell line growth *in vitro*, Y^−^ tumors did grow at a faster rate in immunocompetent mice but not in immunodeficient (Rag^−/−^Il2rg^−/−^) mice, suggesting that the detrimental effects of Y chromosome loss were mediated by the adaptive immunity. The authors also found that the faster tumor growth associated with LOY was associated with *KDM5D* loss and that Y-low bladder cancers had higher infiltration of exhausted and progenitor exhausted T cells [79]. Importantly, this study also showed that Y^−^ tumors are more sensitive to anti-PD-1 immune checkpoint blockers. Y chromosome downregulation, “a male-specific signature of cancer susceptibility”, is thus an important contributor to impaired anti-tumor immunity and, in turn, to the bladder cancer male sex bias. This study also underscores the translational importance of fundamental research. Whether loss of Y gene signatures can be used clinically to guide immunotherapy will have to be determined through prospective clinical trials.

Gene methylation: While the association of LOY and *KDM5D* loss discussed above links sex-specific gene methylation patterns to impaired anti-tumor immunity in men, KDM6A is another epigenetic modifier with likely strong influences in bladder cancer susceptibility by sex. Analyzing a cohort of non-muscle-invasive stage Ta bladder cancers, Hurst et al. reported that 74% of female-derived tumors (compared to 42% of male-derived tumors) contained one or more *KDM6A* mutation [80]. Qiu et al. investigated the possible mechanisms related to KMD6A and reported that KDM6A loss activates cell cycle progression via the transcription factor ATF3 [81]. 

Thus, there are some intriguing associations in our current understanding of sex bias in bladder cancer: AR signaling and T-cell exhaustion, LOY (and *KDM5D* demethylase loss) and T-cell exhaustion, and sex-dependent demethylase (*KDM6A*) gene loss. Kaneko and Li were able to perform a novel study using sex-reversed mice to identify hormonal influences from epigenetic factors in bladder cancer to ultimately conclude that *KDM6A* loss independently contributes to the male sex bias in bladder cancer [82]. The four core genotypes of the sex-reversed mice include “XX and XY males with testes and XX and XY females with ovaries”. Regardless of gonadal sex, XY genotype was associated with lower survival in the BBN mouse model, and *KDM6A* knockout led to shorter survival among wild-type female mice (and similar to XY female mice). Thus, KDM6A has tumor suppressor function, and, as an X-linked gene, it may be an important factor of the male bladder cancer sex bias, which potentially acts in concert with AR/hormonal factors [82].

## 9. Implications for Cancer Prevention and Therapy

Prevention: Given the association of androgen receptor (AR) signaling with bladder carcinogenesis and the likely role of the AR in the substantial male sex bias in bladder cancer, anti-androgen monotherapies may have a future role in bladder cancer prevention. Chemoprevention on the general population level is likely not practical or desirable. However, secondary prevention among individuals at high risk of developing clinically significant bladder cancers (high-risk non-muscle-invasive bladder cancers and invasive bladder cancers) by virtue of prior diagnosis with lower-risk non-muscle-invasive bladder cancers may be feasible. The rationale for such an approach would be to prevent transition from early-stage disease to diagnoses with non-negligible risks of progression and lethality. The contribution of AR signaling to female bladder cancer diagnoses is not well understood; therefore, initial efforts to utilize anti-androgens for secondary bladder cancer prevention will likely initially occur in men.

Therapy: Given the strong associations of AR signaling with impaired anti-tumor immunity via T-cell exhaustion detailed above, combining anti-androgen therapies with immune checkpoint inhibitors may augment responses to systemic immunotherapies. As mentioned above, loss of the Y chromosome has been associated with greater responsiveness to anti-PD-1 immunotherapy [79]. As the majority of bladder cancers are non-invasive and or localized at diagnosis, of equal or greater importance is the potential ability to inhibit f AR signaling to optimize clinical outcomes in conjunction with local (intravesical) immunotherapies such as bacille Calmette–Guérin (BCG), nadofaragene firadenovec [83], nogapendekin alfa inbakicept [84], cretostimogene grenadenorepvec [85], and others. In this specific context, tissue-level biomarkers pertinent to AR signaling and T-cell exhaustion should be evaluated for the prediction of outcomes with a combined anti-AR/immunotherapy treatment approach.

## 10. Conclusions

Androgen receptor (AR) signaling is crucial in bladder cancer pathogenesis. Preclinical studies have shown that AR activates many potential tumor-promoting signaling pathways and appears to substantially alter anti-tumor immunity. Accordingly, suppression of androgen signaling in multiple clinical cohorts has been linked to lower bladder cancer risk and recurrence. Given these compelling associations, targeting AR has been the focus of several ongoing and upcoming clinical trials, which will likely lead to additional valuable insights into the associations of AR and bladder cancer. Key future directions will assess anti-androgens for secondary chemoprevention, as part of a combination approach with bladder-cancer-directed immunotherapies, and whether AR pathway and T-cell functional biomarkers can predict therapeutic outcomes.

## Figures and Tables

**Figure 1 cancers-16-00746-f001:**
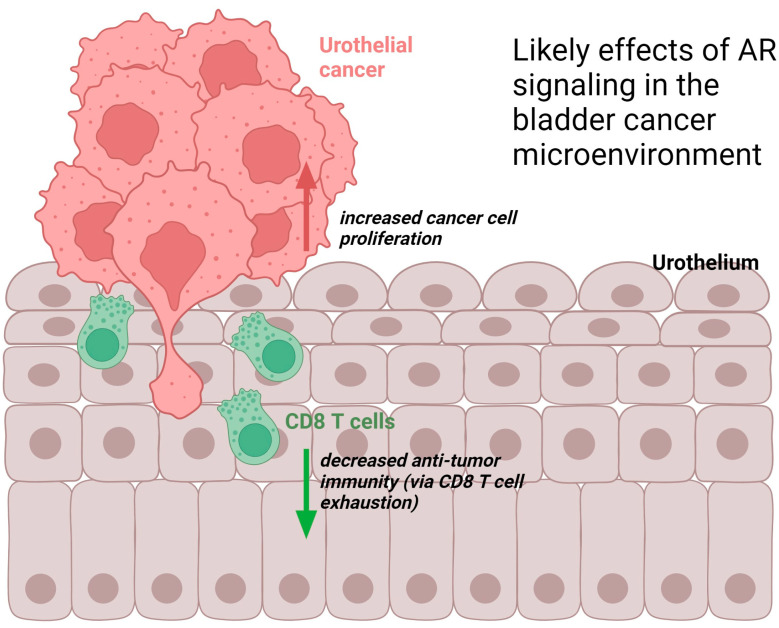
Likely effects of androgen receptor signaling in the bladder cancer microenvironment. Figure created with biorender.com, accessed on 14 December 2023.

**Figure 2 cancers-16-00746-f002:**
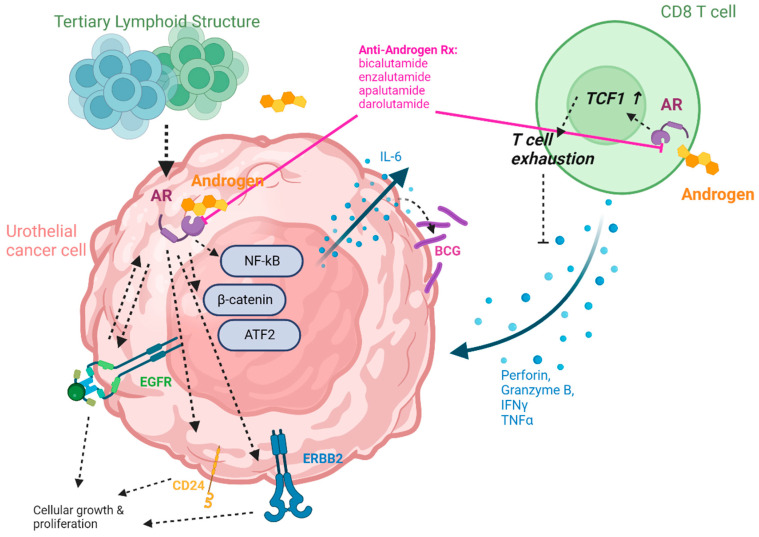
AR signaling pathways relevant to anti-tumor immunity. Figure created with biorender.com.

**Table 1 cancers-16-00746-t001:** Completed and active bladder cancer clinical trials that assess anti-androgen therapies.

NCT Number	Phase	Title	Sample Size	Notes
02300610 [71]	I/Ib	Phase I/Ib Trial of Enzalutamide and Gemcitabine and Cisplatin in Metastatic Bladder Cancer	10 complete	One complete responder had strong tumor AR expression
02605863	II	Enzalutamide for Bladder Cancer Chemoprevention	1 terminated	Low- and intermediate-risk non-muscle-invasive bladder cancer (NMIBC)
04282876		Influence of Hormone Treatment in Radiation Therapy for Bladder Cancer (Observational)	60 recruiting	cT2-4 localized bladder cancers; assessing degarelix association with bladder fibrosis
05327647	II	Phase II Trial of Bicalutamide in Patients Receiving Intravesical BCG for Non-muscle Invasive Bladder Cancer	160 recruiting	BCG versus BCG + bicalutamide
05521698	II	Effects of Apalutamide on EGFR Expression in Patients with Non-muscle Invasive Bladder Cancer	80 estimated start 1/2024	Placebo versus apalatumide in NMIBC
05839119	I	Targeting Androgen Signaling in Urothelial Carcinoma	32 recruiting	Neoadjuvant; subjects with AR + tumors

## Data Availability

All primary data cited in this review are published as noted in the references section and available in the public domain.

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
