# Peer review of "Roles of Androgen Receptor Signaling in Urothelial Carcinoma"

_cancers, 2024, doi:10.3390/cancers16040746_

Round 1

Reviewer 1 Report

Comments and Suggestions for Authors

This is a well written paper, with supporting data and evidence, precepts upon precepts. I have only minor corrections.

The initial sentence is too long, it is a run on sentence-

Bladder cancer is a relatively common solid tumor with a global burden that is amplified by high mortality rates1,2 and intense/lifelong treatment/monitoring requirements leading to the highest treatment costs among all cancer types.

On page 3, this is an uncompleted sentence-

Sathianathen et al. could not confirm the chemopreventive effect of finasteride, however, when conducting a secondary analysis of males in the Medical Therapy for Prostatic Symptoms (MTOPS) trial (which was a much smaller cohort, n=2700, than PLCO).

 What happened when conducting anaylsis of MTOPS? That sentence ended abruptly without a statement of what happened. Then in the following sentence you had a full sentence stating what Sathianathen found. Keep the preceeding sentence and remove the sentence that I have italized in bold.

On page 3, there should be a comma after subsequently,

“Subsequently, Besancon et al. showed in the MBT-2 murine bladder cancer model that

Page 5

Overdevest et al. found that the transmembrane protein CD24, which mediates cell proliferation in several cancer types, is partially responsible for carcinogenesis in the BBN model (especially in male subjects) .

END THE ABOVE SENTENCE HERE. THEN START THE NEXT SENTENCE AS A BRAND NEW SENTENCE. – “Additionally”,CD24 is upregulated in the presence of androgens.

Reviewer 2 Report

Comments and Suggestions for Authors

The review by Sundi et al. summarizes the effect of the androgen receptor and inhibition of its signaling for bladder cancer prevention and therapy.

It is a well written review and important for the general reader. However it should be more critical.

Major points:

1. A very recent review summarizing the same topic appeared in Nature Reviews Urology, doi.org/10.1038/s41585-023-00761-y, which was not cited in this manuscript. Please refer to it and emphasize the distinction to this contribution by Sundi et al.

2. An important finding is that finasteride treatment decreased bladder cancer over time.

Authors discussed well the ongoing clinical trials with AR antagonists. Please include this contribution: doi: 10.3892/ijo.2016.3781

3. Finasteride reduces the production of DHT and is thus rather an indirect AR antagonist.  Please discuss this fact and compare mechanistically with androgen deprivation and with the action of AR antagonists at molecular level.

4. Authors discuss for prevention of bladder cancer anti-androgen monotherapies. Do authors really suggest a population-wide prevention by anti-androgen treatment?

This suggestion should be strongly tuned down since anti-androgens act systemically and have many side-effects. E.g. treatment with anti-androgens may cause selection of aggressive prostate cancer.
